# Aberrant calcium channel splicing drives defects in cortical differentiation in Timothy syndrome

Georgia Panagiotakos[1,2,3†]*, Christos Haveles[2,3†], Arpana Arjun[2,3,4†], Ralitsa Petrova[2,3†], Anshul Rana[5], Thomas Portmann[1‡], Sergiu P Paşca[1,6], Theo D Palmer[7], Ricardo E Dolmetsch[1§]

[1]Department of Neurobiology, Stanford University School of Medicine, Stanford, United States; [2]Eli & Edythe Broad Center of Regeneration Medicine and Stem Cell Research, University of California, San Francisco, San Francisco, United States; [3]Kavli Institute for Fundamental Neuroscience, University of California, San Francisco, San Francisco, United States; [4]Graduate Program in Developmental and Stem Cell Biology, University of California, San Francisco, San Francisco, United States; [5]Graduate Program in Biochemistry, Stanford University School of Medicine, Stanford, United States; [6]Department of Psychiatry and Behavioral Sciences, Stanford University School of Medicine, Stanford, United States; [7]Department of Neurosurgery, Stanford University School of Medicine, Stanford, United States

*For correspondence:
georgia.panagiotakos@ucsf.edu

Present address: [†]Department of Biochemistry and Biophysics, University of California, San Francisco, San Francisco, United States; [‡]Neucyte, Inc, San Carlos, United States; [§]Novartis Institutes for Biomedical Research, Cambridge, United States

**Abstract** The syndromic autism spectrum disorder (ASD) Timothy syndrome (TS) is caused by a point mutation in the alternatively spliced exon 8A of the calcium channel Ca$_v$1.2. Using mouse brain and human induced pluripotent stem cells (iPSCs), we provide evidence that the TS mutation prevents a normal developmental switch in Ca$_v$1.2 exon utilization, resulting in persistent expression of gain-of-function mutant channels during neuronal differentiation. In iPSC models, the TS mutation reduces the abundance of SATB2-expressing cortical projection neurons, leading to excess CTIP2+ neurons. We show that expression of TS-Ca$_v$1.2 channels in the embryonic mouse cortex recapitulates these differentiation defects in a calcium-dependent manner and that *in utero* Ca$_v$1.2 gain-and-loss of function reciprocally regulates the abundance of these neuronal populations. Our findings support the idea that disruption of developmentally regulated calcium channel splicing patterns instructively alters differentiation in the developing cortex, providing important *in vivo* insights into the pathophysiology of a syndromic ASD.

## Introduction

Alterations in intracellular calcium signaling have been linked to neuropsychiatric disease. Genome-wide association studies (GWAS) of patients with bipolar disorder, schizophrenia and autism spectrum disorders (ASD) have identified gain- and loss-of-function mutations in *CACNA1C*, the gene encoding the α1C subunit of the voltage-gated L-type calcium channel (LTC) Ca$_v$1.2 (*Ferreira et al., 2008*; *Green et al., 2010*; *Ripke et al., 2013*). Genetic variants in calcium signaling proteins and voltage-gated calcium channel subunits, including the pore-forming α and auxiliary β subunits of Ca$_v$1.2, have also been associated with disorders such as ASD, schizophrenia, and attention deficit hyperactivity disorder (*Cross-Disorder Group of the Psychiatric Genomics Consortium, 2013*). Although these studies strongly implicate genetic changes in Ca$_v$1.2 in multiple psychiatric disorders, little is known about how Ca$_v$1.2 affects core cellular and molecular processes to contribute to the development of neuropsychiatric disorders.

Timothy syndrome (TS), a developmental disorder characterized by cardiac dysfunction, intellectual disability and ASD, is caused by a gain-of-function missense mutation in an alternatively spliced exon of $Ca_v1.2$ (*Splawski et al., 2004*). The TS mutation, in combination with the use of human induced pluripotent stem cells (iPSCs) from TS patients, has been a powerful tool for investigating the importance of $Ca_v1.2$ signaling in neurodevelopment (*Paşca et al., 2011*; *Krey et al., 2013*; *Tian et al., 2014*; *Birey et al., 2017*). The point mutation in classical TS, G406R, is located in exon 8A, which encodes transmembrane segment 6 (IS6) of channel domain 1. Transcripts containing the G406R mutation encode a channel that exhibits impaired voltage-dependent channel inactivation and, as a result, gives rise to prolonged activity-dependent elevations in intracellular calcium (*Splawski et al., 2004*; *Paşca et al., 2011*; *Krey et al., 2013*).

Immature cells of the embryonic cortex possess a repertoire of ion channels and a chloride resting state that promotes depolarization by GABA (*Ben-Ari, 2002*). Disrupting GABA-induced excitation during this transitional state causes pronounced synaptic and behavioral defects associated with psychiatric disease (*Wang and Kriegstein, 2011*). *In situ* calcium imaging studies during normal development have suggested a role for GABA and glutamate depolarization, as well as synchronous calcium fluctuations, in the proliferation of radially clustered neural progenitor cells (NPCs) in the developing mouse cerebral cortex (*LoTurco et al., 1995*; *Weissman et al., 2004*). More recently, progressive temporally regulated hyperpolarization of cortical NPCs has been linked to the sequential emergence of distinct laminar fates (*Vitali et al., 2018*). The TS mutation in $Ca_v1.2$ has been shown to alter later events in corticogenesis, including the elaboration of dendrites in immature neurons (*Krey et al., 2013*) and the radial migration of upper layer neurons (*Kamijo et al., 2018*), but it remains unclear which channels mediate calcium signals in differentiating cells of the developing cortex and how their activity is controlled during differentiation. Moreover, the effects of these calcium signals on the transcription of downstream factors associated with neuronal specification, as well as their role in coupling electrical activity to extrinsic and intrinsic programs regulating NPC differentiation, are poorly understood.

Cortical development involves both temporal and spatial regulation of NPC differentiation. Upon exiting the cell cycle, newborn excitatory neurons sequentially migrate out of the ventricular and subventricular zones (VZ, SVZ) to their appropriate laminar destination within the cortex (*Okano and Temple, 2009*). During differentiation, these young neurons acquire a number of individual properties that collectively comprise their fate, including patterns of connectivity and electrical activity. The acquisition of neuronal subtype and laminar identity is regulated in part by subtype-specific genetic programs (*Molyneaux et al., 2007*; *Leone et al., 2008*; *Fame et al., 2011*; *Srinivasan et al., 2012*). In layer V, for example, interhemispheric neurons that send projections through the corpus callosum to the contralateral hemisphere (callosal projection neurons, CPNs) are born concurrently with corticofugal neurons that send axons to subcortical structures and the spinal cord (subcerebral projection neurons, SCPNs). This divergent specification is mediated by mutually-repressive transcriptional programs. The specification of SCPNs requires the expression of the transcription factors FEZF2 and CTIP2, whereas persistent expression of the DNA-binding protein SATB2 is a hallmark of callosal projection neuron (CPN) identity (*Arlotta et al., 2005*; *Chen et al., 2005*; *Molyneaux et al., 2005*; *Alcamo et al., 2008*; *Arlotta et al., 2008*; *Britanova et al., 2008*; *Chen et al., 2008*).

Using a human iPSC platform, our laboratory previously generated neurons from patients with TS. Patient cells displayed prolonged intracellular calcium elevations in response to depolarization and deficits in calcium signaling. The resulting changes in gene expression suggested a decrease in CPNs and proportional increase in SCPNs (*Paşca et al., 2011*). Here, using both mouse brain and human iPSC-derived cortical cultures, we show that the differentiation of NPCs into post-mitotic neurons is accompanied by a shift in $Ca_v1.2$ exon utilization from exon 8A to exon 8. iPSC-derived cells from individuals with TS fail to undergo this developmental shift to exon 8 utilization and continue to express gain-of-function channels containing the mutant exon 8A during neuronal differentiation. In a series of *in vivo* experiments in mice, we go on to show that persistent expression of TS gain-of-function channels is alone sufficient to phenocopy the differentiation defects observed in patient-derived neurons, altering the expression of fate determinants during neuronal differentiation in a calcium-dependent manner. Consistent with the idea that altering calcium levels in differentiating NPCs impacts the acquisition of neuronal identity, we also find that *in utero* $Ca_v1.2$ gain- and loss-of-function reciprocally regulate the generation of CPNs and SCPNs. Collectively, these data suggest that the TS mutation gives rise to developmental phenotypes in part by promoting continued expression

of mutant channels that elevate calcium levels in differentiating cells to alter fate specification during corticogenesis. These findings represent the first demonstration of a potential role for electrically evoked signals from LTCs on the regulation of genetic programs specifying neuronal subtype identity in the developing cortex.

## Results

### The Ca$_v$1.2 exon mutated in TS is enriched in undifferentiated progenitors and young neurons in mouse and human cortex

To determine the timing of TS mutant exon expression and the precise developmental window during which the TS mutation exerts its cellular phenotypes, we investigated the expression of *Cacna1c* exon 8A and its mutually exclusive alternate exon 8 during mouse and human corticogenesis. We first measured the relative abundance of exons 8A and 8 using quantitative reverse transcription PCR (qRT-PCR) on RNA extracted from dissected mouse frontal cortices at eight time points spanning embryonic and postnatal corticogenesis (*Figure 1—figure supplement 1a–c*). During corticogenesis, mRNA expression of exon 8A declined after birth, whereas exon 8-containing transcripts increased to levels that plateaued in adulthood (*Figure 1a*). In postnatal cortex, the ratio of exon 8A/exon 8 declined markedly to levels consistent with the significantly higher expression of exon 8 in post-mitotic neurons (*Figure 1—figure supplement 1d*). qRT-PCR on individual cortical cells using Fluidigm dynamic arrays further confirmed that the abundance of exon 8A-containing transcripts decreases per cell to nearly undetectable levels between embryonic day 18 (E18) and postnatal day 14 (P14) (*Figure 1—figure supplement 1e,f*).

We next designed locked nucleic acid (LNA) probes to assess the spatial localization of exon 8- and 8A-containing mRNAs by *in situ* hybridization (*ISH*). Exon 8A-containing mRNAs were expressed in the developing mouse brain as early as E11. At E14, expression of exon 8A transcripts was highest in the VZ/SVZ of the developing cortex and in ventral structures such as the ganglionic eminences and the striatum (*Figure 1b*). Exon 8 mRNA was more broadly expressed in the developing brain (*Figure 1—figure supplement 1g*). Sense controls yielded no signal, as did *ISH* with exon 8 and 8A probes on thymus tissue, which contains no *Cacna1c* transcripts (not shown). At E18, exon 8A expression persisted in the VZs of the dorsal and ventral forebrain (*Figure 1—figure supplement 1h*).

To explore the regulation of these exons in the human brain, we extended these analyses to human fetal cortex and human iPSC neural derivatives. Using previously established protocols (*Paşca et al., 2011*), we differentiated iPSCs from control subjects into dorsal forebrain NPCs and excitatory neurons and observed a dramatic increase in exon 8 expression upon neuronal differentiation, comparable to what we saw in mouse (*Figure 1c*). Similarly, in human fetal cortex, exon 8 expression increased to levels greater than exon 8A as development progressed (*Figure 1—figure supplement 1i*). Together, our data in mouse and human support the idea that exon 8A-containing Ca$_v$1.2 transcripts are enriched in undifferentiated progenitors and young neurons, indicating that the TS mutation may act in immature cells to influence cell fate decisions.

### TS patient cells exhibit altered *CACNA1C* splicing

The G > A nucleotide substitution that causes the TS mutation occurs in the penultimate nucleotide of exon 8A, placing it squarely in position to influence splicing of this exon and its counterpart exon 8. To investigate this possibility, we performed qRT-PCR on TS patient-derived NPCs and neurons. We observed a pronounced splicing defect in patient cells that favored inclusion of the mutant exon 8A in *CACNA1C* transcripts (*Figure 1d–h*). The upregulation of exon 8 that typically occurs with neuronal differentiation was abrogated in TS neurons (*Figure 1d*), and the expression of exon 8A as compared to exon 8 was markedly increased across patient-derived NPC lines (*Figure 1e*). Moreover, the exon 8A/exon 8 ratio in patient cells was significantly increased at the population level (*Figure 1f*, *Figure 1—figure supplement 1j*). Single-cell qRT-PCR using Fluidigm arrays revealed that the proportion of individual neurons expressing exon 8A in TS patients was dramatically increased compared to controls (*Figure 1g*), suggesting continued expression of the TS mutation in a significant fraction of differentiating cells. Notably, the proportion of total neurons in patients and controls was unaltered, as measured by the proportion of cells expressing neural cell adhesion

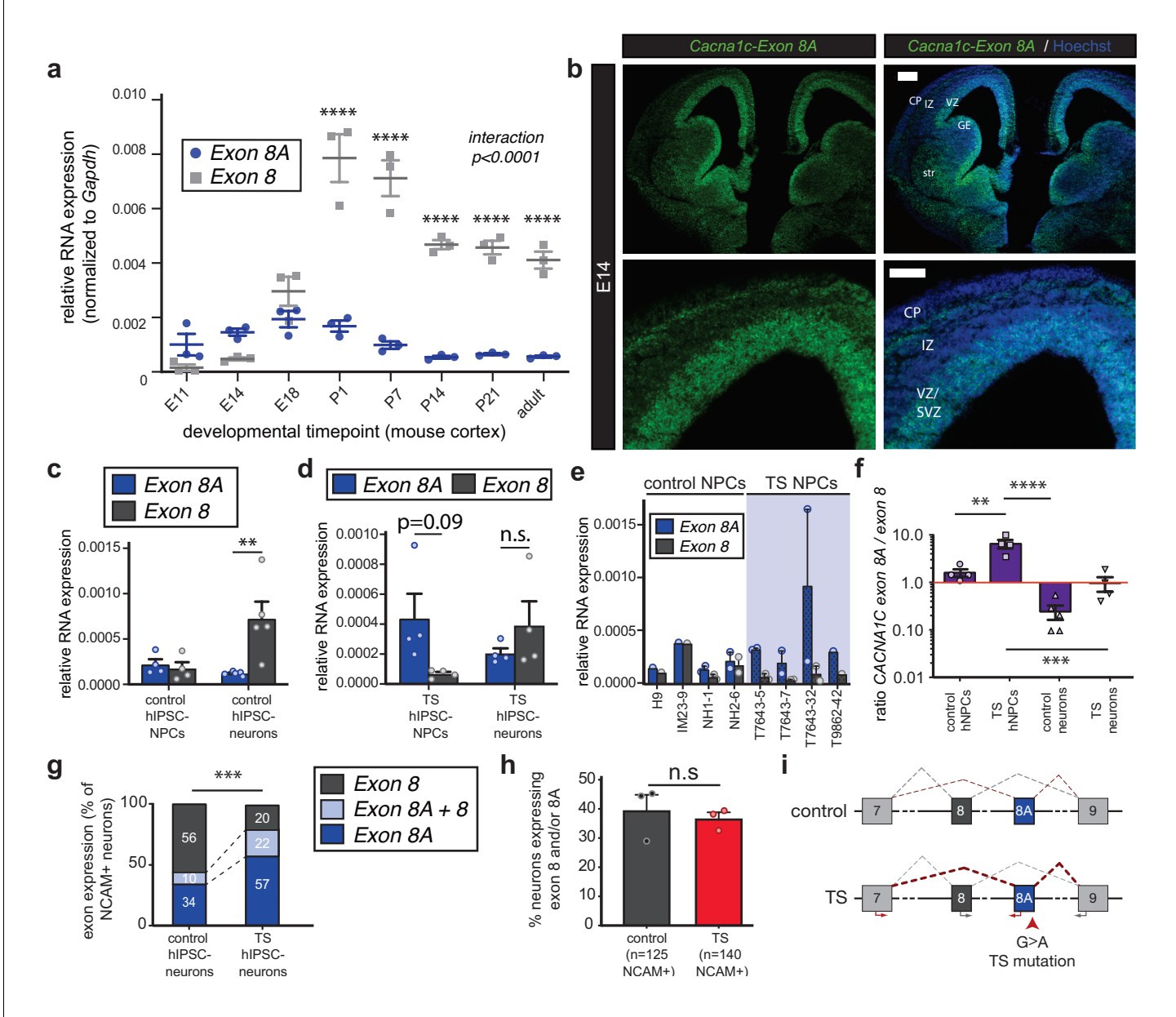

**Figure 1.** The TS mutation alters *CACNA1C* mRNA splicing and induces persistent expression of mutant channels in differentiating human neurons. (**a**) qRT-PCR quantifying relative abundance of *Cacna1c* exons 8 and 8A in developing mouse cortex (n = 3 mice per timepoint from two different litters; data presented as mean ± s.e.m.; ****p<0.0001, two-way ANOVA and post-hoc Bonferroni). (**b**) Representative fluorescence *ISH* images of coronal sections through developing mouse brain at E14 depict strong expression of *Cacna1c*-8A transcripts in neurogenic zones lining the ventricles, in newborn neurons of the CP, and in the developing striatum. VZ, ventricular zone; SVZ, subventricular zone; IZ, intermediate zone; CP, cortical plate; GE, ganglionic eminence; and str, striatum. *Scale bars*, 200 μm (upper), 100 μm (lower). (**c**) qRT-PCR on RNA from differentiating human iPSC-derived NPCs. *CACNA1C*-8 transcripts are upregulated during neuronal differentiation in control cultures (NPCs: two subjects and H9 ES line, four lines total; neurons: two subjects and H9 ES line, five lines total; normalized data (to GAPDH) presented as mean across lines ± s.e.m.; **p<0.005, two-way ANOVA and post-hoc Bonferroni). (**d**) qRT-PCR on RNA from differentiating human TS iPSC-derived NPCs or neurons demonstrates that upregulation of exon 8 is abrogated during neuronal differentiation of TS patient-derived neurons (TS NPCs and neurons: two patients, four lines; T7643-5, T7643-7, T7643-32, and T9862-42; normalized data (to GAPDH) presented as mean across lines ± s.e.m.; n.s., not significant, two-way ANOVA). (**e**) The relative abundance of *CACNA1C* exons 8 and 8A in NPC cultures is shown separately for H9 ES line, three lines from two healthy individuals (IM23-9, NH1-1 and NH2-6), and four TS lines from two individuals (T7643-5, T7643-7, T7643-32, and T9862-42). Data points indicate individual differentiations. In all TS lines examined, exon 8A is more highly expressed than exon 8 in NPCs. (**f**) The ratio of exon 8A to exon 8 decreases in differentiating control neurons, while TS NPCs show an increased exon 8A/exon 8 ratio (data presented as mean ± s.e.m.; **p<0.005, ***p<0.001, ****p<0.0001, one-way ANOVA and post-hoc Bonferroni). (**g, h**) Single-cell qRT-PCR using Fluidigm arrays of neuronal cultures at day 45 of differentiation reveals a greater proportion of neurons

*Figure 1 continued on next page*

*Figure 1 continued*

expressing *CACNA1C*-8A in TS patients compared to controls (**g**), (n = 125 control neurons, n = 140 TS neurons from three control and three patient lines; ***p<0.001, $\chi^2$ = 27.36, Chi-square test). (**h**) The percentage of neurons in patients and controls remains the same, as assessed by NCAM expression (Fluidigm arrays; data presented as mean ± s.e.m., p=0.66, n.s., not significant, unpaired t-test). (**i**) A working model depicting the splicing shift caused by the TS mutation in a schematized version of the *CACNA1C* genomic locus spanning exons 7 to 9.

The online version of this article includes the following source data and figure supplement(s) for figure 1:

**Source data 1.** Expression of exons 8 and 8A in the mouse cortex and differentiating human IPSCs.
**Figure supplement 1.** *Cacna1c* exon 8 and 8A expression in mouse and human.
**Figure supplement 1—source data 1.** qRT-PCR and single cell qRT-PCR quality controls and raw data.

molecule (NCAM) (*Figure 1h*). These data suggest that the TS mutation in exon 8A favors the inclusion of exon 8A in *CACNA1C* transcripts at the expense of exon 8 expression (*Figure 1i*).

## Mimicking TS with *in utero* Ca$_v$1.2 gain of function alters the relative abundance of CPNs and SCPNs in the developing cortex in a calcium-dependent manner

Consistent with our RNA expression analyses, immunofluorescence staining for Ca$_v$1.2 in embryonic mouse cortex demonstrated weak expression of Ca$_v$1.2 channels in NPCs lining the ventricles with significantly higher levels in neurons in the cortical plate (CP), as early as E11 (*Figure 2a*, *Figure 2—figure supplement 1a*). Within the VZ, Ca$_v$1.2 expression was not uniform across cells (*Figure 2b*), as expression of the channel was restricted to a subset of NPCs early in corticogenesis and subsequently spread to a majority of cells by E16. Late in corticogenesis, Ca$_v$1.2 levels increased dramatically (*Figure 2c*), likely owing to elevated expression of the channel in post-mitotic neuroblasts and neurons. To test whether the Ca$_v$1.2 channels expressed in immature cells of the developing cortex contribute to intracellular calcium elevations in response to depolarization, we performed ratiometric calcium imaging using Fura-2 following application of GABA. For this analysis, we dissociated E12.5-E14.5 wild-type mouse cortices and grew cells on coverslips in the presence of bFGF and EGF (20 ng/ml) for 6–24 hr. We observed calcium rises in NPCs and immature neurons induced by depolarization with 30 μM GABA (*Figure 2—figure supplement 1b–d*) in a fraction of cells (20–55% in our experiments), in line with previous studies using GABA to depolarize NPCs in slices of embryonic cortex (*LoTurco et al., 1995*). GABA-induced calcium rises were nearly completely abrogated by the application of 5 μM nimodipine, a selective LTC blocker (*Figure 2—figure supplement 1c,d*), indicating that LTCs control calcium signals in NPCs and young neurons in the developing cortex. Extending these analyses to human, we found that Ca$_v$1.2 protein was robustly expressed throughout the cerebral wall in gestational week (GW) 16 human cortex, with lower expression in the VZ and inner SVZ (*Figure 2—figure supplement 1e*), similar to our results in mouse. Of note, Ca$_v$1.2 was strongly expressed in the outer SVZ (oSVZ, *Figure 2—figure supplement 1e*), an expanded neurogenic zone in the primate lineage containing newly identified classes of progenitor cells (*Fietz et al., 2010*; *Hansen et al., 2010*). This finding suggests a continuing role for this channel in the maintenance and/or differentiation of these progenitor cells. Together, our observations in mouse and human cortex indicate that Ca$_v$1.2 is expressed at the right time and place to influence the process of neuronal differentiation and subtype specification.

In light of the splicing defects observed in patient neurons, we next directly examined the role of Ca$_v$1.2 on differentiation using a series of *in utero* gain-of-function manipulations in mouse. We generated and validated Ca$_v$1.2 *in vivo* expression constructs containing exon 8 with a downstream IRES-EGFP sequence (pCAGIG-Ca$_v$1.2, *Figure 2d,f*, *Figure 2—figure supplement 1f,g*). Consistent with previous work demonstrating that the TS mutation prolongs voltage-dependent channel inactivation and depolarization-induced calcium rises (*Splawski et al., 2004*; *Paşca et al., 2011*), calcium elevations resulting from expression of TS channels in Neuro2A cells were more sustained as compared to expression of channels without the mutation (*Figure 2d*). We introduced Ca$_v$1.2-TS and control IRES-EGFP plasmids into wild-type pregnant mice at E13.5 (*Figure 2e*), corresponding to the birth of layer V CPN and SCPN projection neurons. We sacrificed dams at E17.5, at which point layer V is mostly formed and a majority of electroporated cells had reached the CP (*Figure 2—figure supplement 1h*). We then quantified the relative proportion of EGFP+ cells expressing the transcription

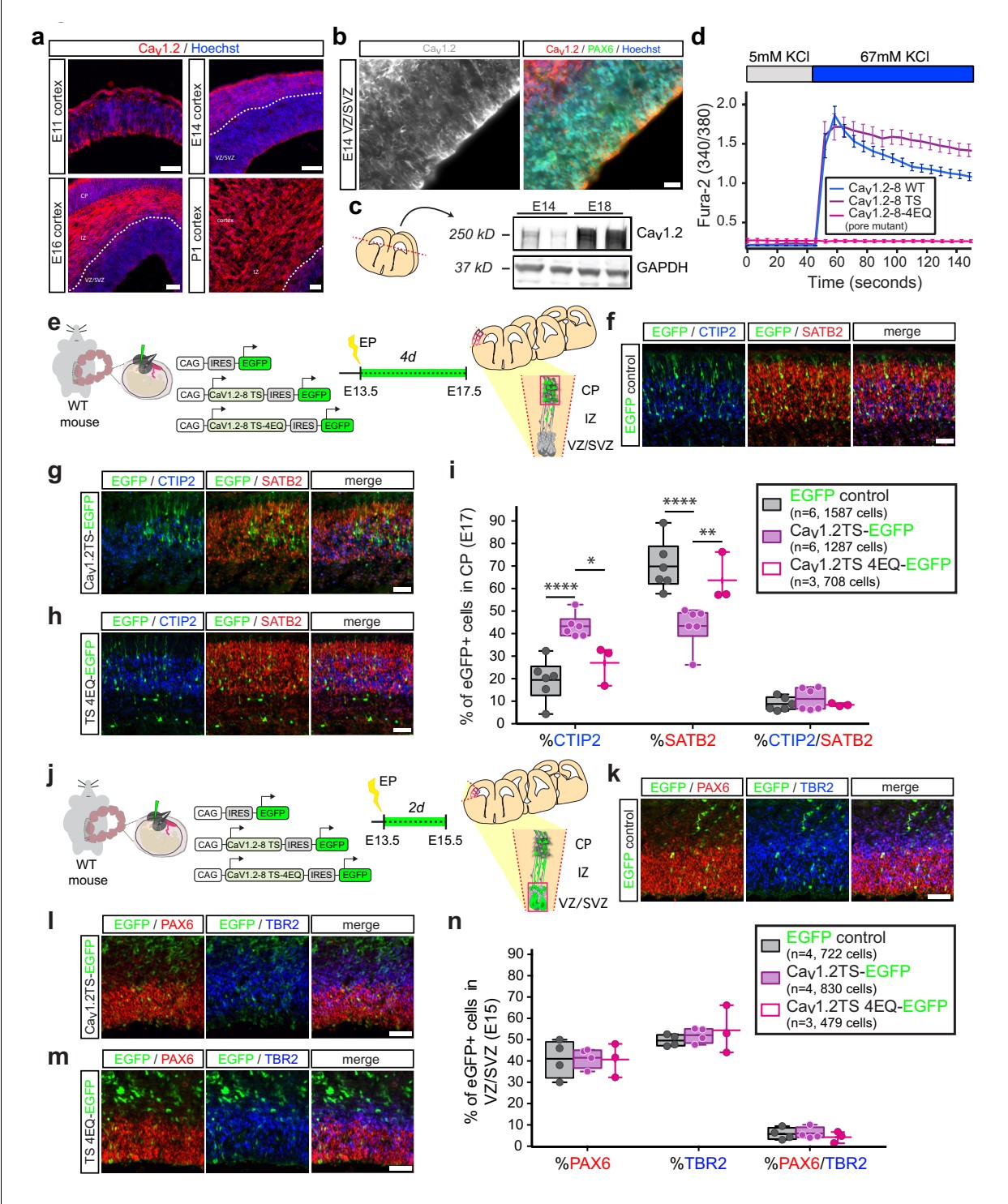

**Figure 2.** *In utero* overexpression of Ca$_v$1.2-TS channels reduces SATB2-expressing cells and increases CTIP2+ cells in the CP in a calcium-dependent manner. (**a**) Representative immunostained coronal sections through the mouse cortex at E11, E14, E16, and P1 depict robust expression of Ca$_v$1.2 (red). Dashed white line delineates border between VZ/SVZ and IZ. VZ, ventricular zone; SVZ, subventricular zone; IZ, intermediate zone; and CP, cortical plate. *Scale bar*, 50 μm. (**b**) (left) High magnification image through mouse VZ depicts Ca$_v$1.2 immunostaining in gray. (right) Ca$_v$1.2 (red) is expressed in Pax6 (green)-positive NPCs. *Scale bar*, 20 μm. (**c**) Western blot of embryonic mouse cortical lysates probed with Ca$_v$1.2 and GAPDH-specific antibodies at E14 and E18 depicts increasing Ca$_v$1.2 levels during corticogenesis. (**d**) Ratiometric calcium imaging of Fura-2-loaded Neuro2A neuroblastoma cells transfected with *in utero* expression constructs. Average calcium response traces are shown for wild-type, TS, and pore mutant channels upon depolarization with 67 mM KCl. Note that pore mutant channels result in a completely abrogated calcium response and, consistent with

*Figure 2 continued on next page*

Figure 2 continued

loss of voltage-dependent inactivation, the TS channel causes a persistent elevated calcium rise. (e) Schematic depicting time course, expression vectors, and mediolateral placement of counter windows for *in utero* $Ca_v1.2$-TS gain-of-function experiments in f–i). (f–h) Representative cropped coronal sections through the CP of electroporated embryos immunostained for EGFP (green), SATB2 (red) and CTIP2 (blue) at E17.5. *Scale bar*, 50 µm. (i) Introducing $Ca_v1.2$-TS *in utero* results in calcium-dependent alterations in the proportion of SATB2- and CTIP2-expressing EGFP+ cells. (IRES-GFP, n = 6 mice, 1587 cells; $Ca_v1.2$-TS, n = 6 mice, 1287 cells; $Ca_v1.2$-TS-4EQ, n = 3 mice, 708 cells; data presented as box and whisker plot, box bounds the interquartile range (IQR) divided by the mean and whiskers extend to the minimum and maximum value; *p<0.05, **p<0.005, ****p<0.0001, two-way ANOVA and post-hoc Bonferroni.) (j) Schematic illustration depicting time course, expression vectors, and mediolateral placement of counter windows for *in utero* $Ca_v1.2$-TS gain-of-function experiments in k–n). (k–m) Representative coronal sections through the VZ/SVZ of electroporated embryos immunostained for EGFP (green), PAX6 (red) and TBR2 (blue) at E15.5. *Scale bar*, 50 µm. (n) Introducing $Ca_v1.2$-TS *in utero* does not alter the distribution of NPC subtypes in the VZ/SVZ (IRES-GFP, n = 4 mice, 722 cells; Cav1.2-TS, n = 4 mice, 830 cells; Cav1.2-TS-4EQ, n = 3 mice, 479 cells; as above, data presented as box and whisker plot; n.s., not significant, two-way ANOVA and post-hoc Bonferroni.).

The online version of this article includes the following source data and figure supplement(s) for figure 2:

**Source data 1.** $Ca_v1.2$-TS *in utero* electroporation experiments.
**Figure supplement 1.** $Ca_v1.2$-TS *in utero* electroporation experiments.

factors CTIP2 and SATB2 in the CP. $Ca_v1.2$-TS electroporation dramatically reduced the abundance of SATB2-expressing EGFP+ cells and increased CTIP2 expression, consistent with our previous gene expression findings in human TS-derived neurons (*Paşca et al., 2011*) (*Figure 2f–g and i*, *Figure 2—figure supplement 1i*). Importantly, electroporation of calcium-impermeable $Ca_v1.2$-TS-4EQ mutant channels did not alter the relative abundance of CTIP2- and SATB2-expressing populations (*Figure 2d,h–i*, *Figure 2—figure supplement 1i*), indicating that calcium influx through $Ca_v1.2$ is required for the differentiation defects observed with expression of the TS mutation. To rule out the possibility that these differentiation defects resulted from earlier $Ca_v1.2$-dependent alterations of the NPC pool, we electroporated control IRES-GFP, $Ca_v1.2$-TS, and $Ca_v1.2$-TS-4EQ constructs into wild-type pregnant mice at E13.5 and quantified the proportion of PAX6+ radial glia and TRB2+ intermediate progenitors in the VZ/SVZ at E15.5 (*Figure 2j*, *Figure 2—figure supplement 1j*). We observed no alterations in the relative abundance of progenitor types with these manipulations, suggesting that calcium through $Ca_v1.2$ impacts the onset of determinants of cortical neuron projection identity in differentiating NPCs (*Figure 2k–n*, *Figure 2—figure supplement 1j*).

Collectively, our observations thus far suggest that modulating calcium levels in differentiating cells plays a role in the choice between adoption of CTIP2 or SATB2 expression in the developing cortex. In line with this idea, *in utero* gain of function of $Ca_v1.2$-WT and $Ca_v1.2$-WT-4EQ channels at E13.5 also resulted in a calcium-dependent reduction in SATB2-expressing CPNs in the cortical plate at E17.5 and a commensurate increase in CTIP2+ SCPNs (*Figure 3a–d*, *Figure 3—figure supplement 1a*). As with expression of $Ca_v1.2$-TS channels, we also assessed NPC distribution in the VZ/SVZ among electroporated cells and found no changes in the relative proportion of radial glia and intermediate progenitor populations that give rise to immature neurons (*Figure 3e–h*, *Figure 3—figure supplement 1b*).

## $Ca_v1.2$ loss of function increases SATB2 expression in the developing cortex

As the observed reduction in putative CPNs resulting from $Ca_v1.2$ gain of function was dependent on calcium influx, we posited that $Ca_v1.2$ loss of function may also impair differentiation. To interrogate the effect of $Ca_v1.2$ deletion on differentiation, we employed an intersectional strategy pairing *in utero* electroporation of Cre recombinase-expressing constructs with mice bearing one null (*Cacna1c* $^-$) and one conditional allele (*Cacna1c*$^{flx}$) of $Ca_v1.2$.

We electroporated Cre:EGFP fusion constructs into E12.5 embryos resulting from *Cacna1c*$^{+/-}$ and *Cacna1c*$^{flx/+}$ parent crosses, in order to ensure recombination of the conditional allele by E13, when layer V neurons are generated. Similar to our previous analyses, we quantified electroporated cells 4 days following the electroporation, at E16.5. As with the $Ca_v1.2$ gain-of-function experiments, we quantified the relative abundance of electroporated cells expressing the fate determinants CTIP2 or SATB2 in the CP (*Figure 4a–c*). Loss of only one copy of $Ca_v1.2$ (*Cacna1c*$^{+/-}$) resulted in a significant reduction in the proportion of CTIP2-expressing EGFP+ cells and an increase in SATB2-expressing cells in the CP (*Figure 4d*, *Figure 4—figure supplement 1a*), opposite to what we found with TS

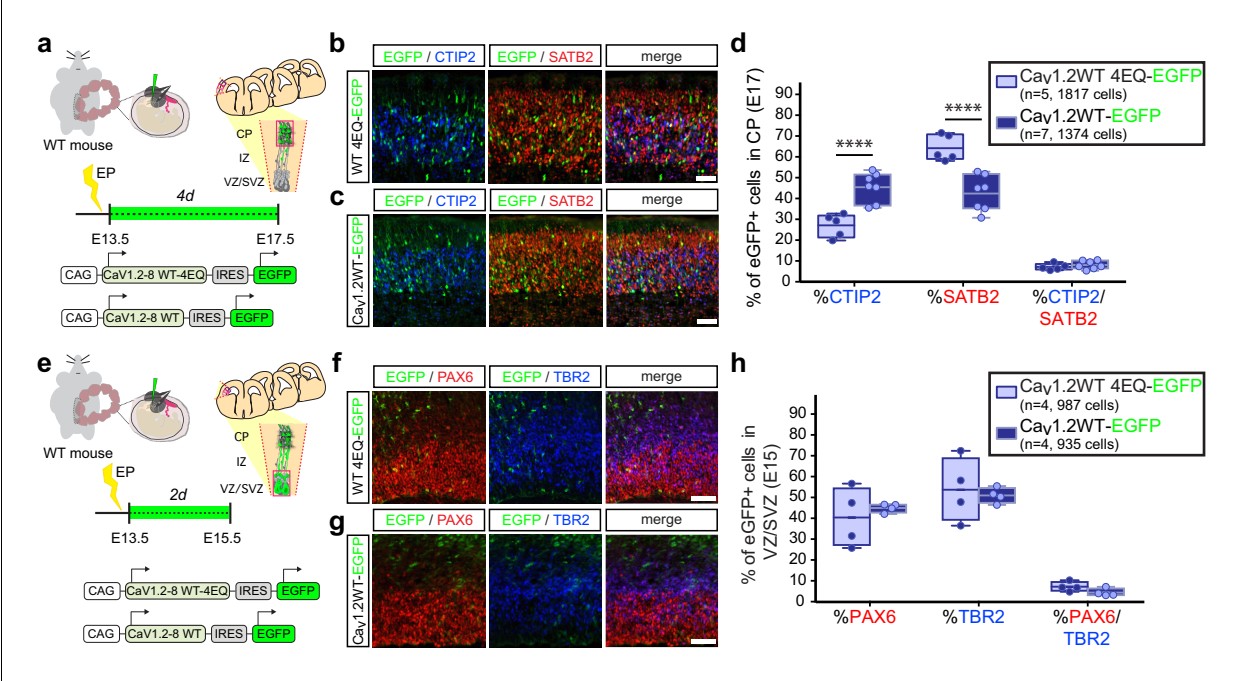

**Figure 3.** *In utero* over expression of wild-type Ca$_v$1.2 alters the relative abundance of early-born projection neuron subtypes similar to TS channels. (a) Schematic illustration depicting time course, expression vectors, and mediolateral placement of counter windows for *in utero* Ca$_v$1.2-WT gain-of-function experiments in b–d). (b, c) Representative coronal sections through the CP of electroporated embryos immunostained for EGFP (green), SATB2 (red) and CTIP2 (blue) at E17.5. *Scale bar*, 50 μm. (d) Introducing Ca$_v$1.2-WT *in utero* results in a calcium-dependent increase in the relative abundance of SATB2-expressing EGFP+ cells and a commensurate reduction in CTIP2+ cells (Ca$_v$1.2-WT-4EQ, n = 5 mice, 1817 cells; Ca$_v$1.2-WT, n = 7 mice, 1374 cells; data presented as box and whisker plot, box bounds the interquartile range (IQR) divided by the mean and whiskers extend to the minimum and maximum value; ****p<0.0001, two-way ANOVA and post-hoc Bonferroni.) (e) Schematic illustration depicting time course, expression vectors, and mediolateral placement of counter windows for *in utero* Ca$_v$1.2-WT gain-of-function experiments in f–h). (f,g) Representative coronal sections through the VZ/SVZ of electroporated embryos immunostained for EGFP (green), PAX6 (red) and TBR2 (blue) at E15.5. *Scale bar*, 50 μm. (h) *In utero* expression of Ca$_v$1.2-WT does not alter the relative abundance of NPC subtypes in the VZ/SVZ (Ca$_v$1.2-WT-4EQ, n = 4 mice, 987 cells; Ca$_v$1.2-WT, n = 4 mice, 935 cells; as above, data presented as box and whisker plot; n.s., not significant, two-way ANOVA and post-hoc Bonferroni.)

The online version of this article includes the following source data and figure supplement(s) for figure 3:

**Source data 1.** Ca$_v$1.2-WT *in utero* gain-of-function experiments.
**Figure supplement 1.** Ca$_v$1.2 gain-of-function *in utero* electroporation experiments.

channels and Ca$_v$1.2 gain of function. We observed no additional enhancement of this effect in the floxed/null condition, likely resulting from persistence of Ca$_v$1.2 protein. In addition to the observed effects on CTIP2 and SATB2, Ca$_v$1.2 loss also resulted in decreased expression of TBR1 (*Figure 4e–f*), a transcription factor required for differentiation of layer VI cortico-thalamic projection neurons that is also expressed at lower levels downstream of SATB2 in upper layer neurons (*Srinivasan et al., 2012*; *Hevner et al., 2001*). This result is consistent with previous studies in adult brain suggesting that bicuculine- or glutamate-induced increases in excitation upregulate TBR1 expression (*Chuang et al., 2014*).

## Discussion

Two key findings of this study are that the splicing of exons 8 and 8A of the *CACNA1C* gene, which encodes the pore-forming subunit of Ca$_v$1.2, is developmentally regulated in the mouse and human and that the TS mutation in exon 8A alters this regulation. Recent work has described significant *CACNA1C* transcript diversity in the adult human brain (*Clark et al., 2020*), and previous studies have suggested temporal regulation of the splicing of some exons of Ca$_v$1.2 in both the heart and brain (*Diebold et al., 1992*; *Tang et al., 2009*; *Tang et al., 2011*). For example, the inclusion of exons 9* and 33 in Ca$_v$1.2 transcripts, which confer distinct physiological channel properties, is

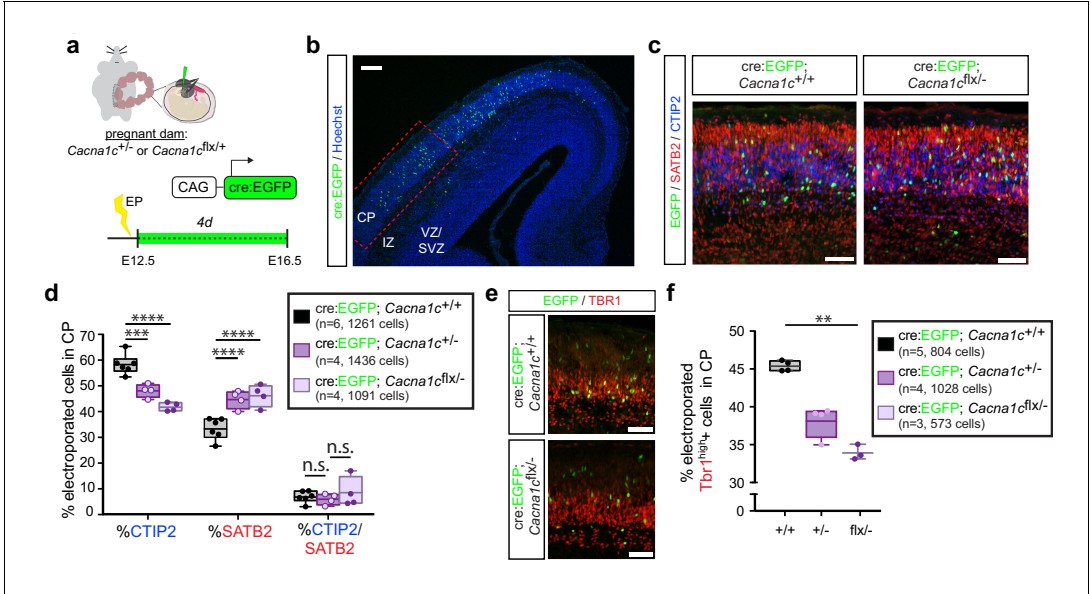

**Figure 4.** Ca$_v$1.2 loss of function results in increased SATB2+ cells and decreased CTIP2+ cells in the CP. (a) Schematic depicting the time course of *in utero* loss-of-function experiments. (b) Expression of cell-type-specific transcription factors was quantified in EGFP+ cells in the CP of E16.5 embryos electroporated with Cre:EGFP at E12.5, in the region marked by the dotted red box. VZ, ventricular zone; SVZ, subventricular zone; IZ, intermediate zone; and CP, cortical plate. *Scale bar*, 100 μm. (c) Representative coronal sections immunostained for EGFP (green), SATB2 (red) and CTIP2 (blue) through the CP of WT and *Cacna1c*$^{flx/-}$ -electroporated embryos at E16.5. *Scale bar*, 50 μm. (d) Ca$_v$1.2 loss of function results in reduced CTIP2-expressing EGFP+ cells with a concomitant increase in SATB2+ cells. (*Cacna1c*$^{+/+}$, n = 6 mice, 1261 cells; *Cacna1c*$^{+/-}$, n = 4 mice, 1436 cells; *Cacna1c*$^{flx/-}$, n = 4 mice, 1091 cells; data presented as box and whisker plot, box bounds the IQR divided by the mean and whiskers extend to the minimum and maximum value; ****p<0.0001, two-way ANOVA and post-hoc Bonferroni.) (e) Representative coronal sections immunostained for EGFP (green) and TBR1 (red). Scale bar, 50 μm. (f) Ca$_v$1.2 loss of function results in a modest reduction in TBR1+ electroporated cells. (*Cacna1c*$^{+/+}$, n = 5 mice, 804 cells; *Cacna1c*$^{+/-}$, n = 4 mice, 1028 cells; *Cacna1c*$^{flx/-}$, n = 3 mice, 573 cells; data presented as box and whisker plot as in (d); **p<0.01, Kruskal-Wallis test and post-hoc Dunn's correction, mean rank compared to Ca$_v$$^{+/+}$.)

The online version of this article includes the following source data and figure supplement(s) for figure 4:

**Source data 1.** Ca$_v$1.2 loss-of-function experiments.
**Figure supplement 1.** Ca$_v$1.2 loss-of-function *in utero* electroporation experiments.

inversely regulated during embryonic brain development (*Tang et al., 2009*). In smooth muscle and heart, transcript scanning has also revealed tissue-specific splicing preferences, suggesting that precise utilization of exons is influenced by combinatorial and coordinated splicing mechanisms (*Tang et al., 2007*). Here, we found that exons 8 and 8A in the *CACNA1C* gene are dynamically regulated during mouse and human cortical development, consistent with a previous study in the mouse identifying an exon 8 splice repressor (*Tang et al., 2011*). These exons are expressed in a mutually exclusive fashion, and we show in this study that exon 8A is expressed at lower levels predominantly in undifferentiated cells early in development, whereas expression of exon 8 increases as neurons mature. How other exons may be coordinately regulated alongside exons 8 and 8A during neuronal differentiation remains unclear. Aside from combinatorial splicing of the pore-forming subunit, channel function can also be modulated by interactions with different auxiliary subunits and their splice isoforms. Therefore, a comprehensive analysis of full-length Ca$_v$1.2 isoform expression in different classes of NPCs and neurons is an essential next step towards understanding the complex cell-type-specific regulation of calcium signaling during differentiation.

Previous studies investigating human Ca$_v$1.2 variants differing only in the inclusion of exon 8 or 8A report that exchanging these two exons yields channel isoforms with similar electrophysiological properties and only subtle changes in sensitivity to dihydropyridine calcium channel inhibitors (*Welling et al., 1997*; *Zühlke et al., 1998*). By contrast, the TS mutation causes a pronounced loss of voltage-dependent inactivation (VDI) when introduced into either exon 8 or 8A. This results in persistent calcium influx through Ca$_v$1.2 channels upon depolarization and accompanying deficits in calcium signaling and activity-dependent gene expression (*Splawski et al., 2004*; *Paşca et al., 2011*).

Here, we show that the TS mutation changes Ca$_v$1.2 splicing by inducing a shift that favors extended use of exon 8A in patient cells. The TS mutation thus not only increases calcium influx by impairing VDI, but also, by promoting continued expression of exon 8A, it ensures that gain-of-function channels containing the TS mutation continue to be expressed during differentiation. This change in splicing likely results from the fact that the TS mutation lies in the splice acceptor site of exon 8A, making it favored relative to the splice acceptor site of exon 8. Continued inclusion of exon 8A also prevents the change in *CACNA1C* splicing that would lead to expression of the wild-type channel at the appropriate time in development. Failure of this developmental splicing switch in TS alters the relative abundance of channels containing WT and mutant exons during development to contribute to cellular phenotypes caused by the mutation. While the full mechanistic impact of this alteration in splicing needs further investigation, the developmental difference in exon utilization reported here could explain the phenotypic differences observed between patients with the TS mutation in exon 8A and those in exon 8.

A significant mechanism underlying the gain-of-function effects of the TS mutation on differentiation appears to be due to alterations in exon use. As the TS mutation is known to prolong channel inactivation and thus intracellular calcium rises in response to depolarization, we reasoned that persistent expression of the TS mutation, owing to extended inclusion of exon 8A, would likely result in elevated activity-dependent calcium signaling as immature cells transition from NPC to post-mitotic neuron. We further hypothesized that these altered calcium dynamics could give rise to the differentiation defects observed in TS. It is becoming increasingly evident that neurons with distinct gene expression profiles, physiological properties, and projection patterns can be generated concomitantly during development and ultimately inhabit the same cortical layer. This observation is best exemplified in layer V, where a complex network of mutually repressive transcription factors sets the stage for the generation of SATB2-expressing CPNs and CTIP2-expressing SCPNs. It is not fully understood, however, how other intrinsic and extrinsic factors interact with these genetic programs to influence the specification of neuronal identity in the developing cortex. Here we show that introducing TS-Ca$_v$1.2 calcium channels into mouse NPCs *in utero* reduces the fraction of SATB2-expressing cells, altering the ratio of early-born cortico-cortico CPNs relative to subcortically projecting SCPN neurons in the developing cortex without changing the abundance of progenitor subtypes that give rise to these cells. Our calcium imaging data, consistent with previous studies, confirms that the TS-Ca$_v$1.2 channels we introduced *in utero* result in sustained calcium elevations as compared to channels that do not express the TS mutation. Collectively, the suppression of SATB2 identity resulting from expression of these TS-Ca$_v$1.2 channels, together with the finding that channels that do not carry calcium do not affect the fraction of SATB2-expressing neurons, support the idea that intracellular calcium levels influence the expression of neuronal fate determinants in the developing cortex. Moreover, these data suggest that persistent elevations in calcium resulting from aberrant channel splicing and continued expression of mutant channels may underlie the defects in cortical differentiation seen in TS.

The idea that calcium may play an important role in the differentiation of cortical neurons is further supported by the Ca$_v$1.2 gain- and loss-of-function studies in our study and is consistent with our previous work demonstrating that human iPSC-derived TS cortical neurons have reduced expression of SATB2 and increased expression of CTIP2 (*Paşca et al., 2011*; *Krey et al., 2013*), in addition to deregulated calcium signaling. It should be noted here, however, that our expression and calcium imaging data suggest that Ca$_v$1.2 protein is expressed at relatively low levels in the ventricular zone and in only a fraction of NPCs. This is a striking contrast to neurons, in which Ca$_v$1.2 protein is more robustly expressed. We suspect, based on how long it would likely take to express functional channels at the cell membrane in our *in utero* electroporation experiments, that Ca$_v$1.2 gain of function primarily occurs as NPCs exit the cell cycle and initiate differentiation. Nevertheless, we cannot exclude the possibility that part of the gain of function phenotype resulting from electroporation of wild-type Ca$_v$1.2 channels is caused by expressing Ca$_v$1.2 at higher levels in cells that normally express it at relatively low levels, or, alternatively, not at all. Despite this possibility, our findings collectively reinforce the idea that elevated calcium is likely behind the changes we see in the relative abundance of CTIP2- and SATB2-expressing populations.

Precisely how calcium signaling may regulate the fate of developing cortical neurons remains a mystery. It is possible that CTIP2, SATB2, or other upstream regulators of projection neuron fate choice, such as the transcription factor FEZF2, are modulated via calcium. FEZF2 expression, for

example, can be induced by brief neuronal activation (*Tyssowski et al., 2018*). In differentiating keratinocytes, CTIP2 degradation appears to be dependent on intracellular calcium levels (*Zhang et al., 2012a*), whereas in T cell fate commitment, MAPK signaling and sumoylation regulate CTIP2 activity in a coordinated fashion (*Zhang et al., 2012b*). Alternatively, calcium signaling may regulate the activity of subunits of neuronal chromatin remodeling complexes, including neuronal BAF (mSWI/SNF) complexes (*Aizawa, 2004*), with which CTIP2 itself associates. Moving forward, it will be essential to establish the precise timing and influence of electrical activity and calcium signaling throughout the process of differentiation, and how electrical activity relates to mechanisms that underlie the effects of $Ca_v1.2$ on cell fate decisions in normal and dysfunctional corticogenesis.

We propose that the changes in cortical excitatory neuron differentiation that we observe upon expression of TS channels significantly contribute to the neurodevelopmental phenotypes associated with TS. It is tempting to speculate that one way in which altered channel splicing and persistent expression of the G406R mutation cause ASD and intellectual disability in TS patients is by altering the ratio of cortico-cortico and subcerebral projection neurons in the cortex. This is unlikely to be the only mechanism by which the TS mutation contributes to ASD phenotypes, as other studies have demonstrated significant effects of the mutation on later developmental processes in the cortex like dendritic arborization and neuronal migration (*Krey et al., 2013*; *Kamijo et al., 2018*), which are also likely to affect circuits that control learning and social behavior in patients. Our *in situ* hybridization experiments suggest that cells outside the developing cortex, such as striatal projection neurons, express exon 8A during embryonic development, as do immature cells in the germinal zones that give rise to cortical interneuron populations and nuclei of the basal ganglia. This observation indicates that the early development of other cell types is also likely to be influenced by the TS mutation. Indeed, using an iPSC spheroid platform, we have shown that cortical interneurons derived from TS patient cells display abnormal migration (*Birey et al., 2017*). It would be interesting to interrogate how aberrant splicing induced by the TS mutation might contribute to this altered migration and to cellular phenotypes in other neuronal populations. Our study provides evidence that calcium influences the differentiation of excitatory neurons in the cortex and that the expression of $Ca_v1.2$ channels containing the TS mutation in differentiating cells may underlie changes in neuronal circuitry that contribute to intellectual disability and ASD in TS patients.

## Materials and methods

### Tissue preparation

All animal experiments were approved by the Stanford University and UCSF Institutional Animal Care and Use Committees and conducted in accordance with the Stanford University, UCSF and National Institutes of Health guidelines for the care and use of laboratory animals. For region-specific qRT-PCR analyses in the mouse, embryonic and postnatal C57/bl6 brains were carefully extracted and microdissected using two syringes and immediately processed for RNA isolation using the Trizol method. Three animals from two different litters were utilized for each developmental timepoint. Brains were then extracted and processed for RNA as described below. For human developing cortex qRT-PCR, RNA from individual frontal cortices at different developmental timepoints was obtained from Biochain and processed as below.

For immunofluorescence staining, embryonic brains were carefully extracted and placed at 4°C in 4% paraformaldehyde for 30 min, 15% sucrose for one hour, and 30% sucrose overnight until embedding. Brains were then cryo-protected in 30% sucrose until embedding in dry ice. Embedding was performed in Optimal Cutting Temperature (O.C.T) and 10–20 µm sections were cut on a freezing cryostat. Sections were stored at −80°C until used for immunohistochemical analyses. Human GW16-17.5 cortical tissue specimens were a gift from Steven Sloan and Dr. Ben Barres. Tissue was post-fixed overnight in 4% paraformaldehyde, cryo-protected in 30% sucrose until embedding and subsequently sectioned on a freezing cryostat at 16 µm. Sections were stored at −80°C until use for immunofluorescence staining.

## *In situ* hybridization

Locked nucleic acid probes directed against 20-nucleotide sequences spanning the 5' regions of *Cacna1c* exons 8 and 8A were generated (Exiqon). The probe sequences are included below, with the + signs indicating the locations of LNA bases:

exon 8: 5'-biotin-A+CT CA+T AGC +CCA TAG CGT +CT-3'
exon 8A: 5'-dig-AGT+CC+CTTCGTA+CGGCATCA-3'
sense probe: 5'-biotin-TGCGA+TA+CC+CGATA+CT+CAAC-3'

Thawed 12–16 µm sections were briefly post-fixed in fresh RNAase free 4% paraformaldehyde for 10 min at 25°C, followed by two washes in DEPC-H$_2$O for 10 min each. Slides were then acetylated for 5 min in 0.1M triethanolamine in HCl (pH 7.5) with drop wise addition of acetic anhydride immediately prior to use. Slides were then washed on a shaker two times for 3 min each in DEPC-PBS. *Prehybridization*: Slides were briefly pre-hybridized in hybridization buffer for 30 min. Hybridization buffer consisted of 50% vol/vol formamide, 5X SSC, 500 µg/µl yeast tRNA, and 1x Denhardt's solution in DEPC-treated H$_2$O. *Hybridization*: Sections were then hybridized overnight in the same hybridization solution containing heat-denatured dig- or biotin-labeled LNA probes. On the following day, stringency washes were performed three times for 10 min each on a shaker in 0.1x SSC at a temperature 4–8°C higher than the hybridization temperature. Sections were then washed once for 5 min in 2x SSC, followed by immunological detection of dig or biotin labeling. Slides were treated with freshly made 3% H$_2$O$_2$ for 10 min to inactivate endogenous peroxidases. Following three washes for 3 min each with TN buffer (0.1M Tris-HCl, pH 7.5; 0.15M NaCl) on a shaker, slides were blocked for 30 min in blocking buffer (0.1M Tris-HCl, pH 7.5; 0.15M NaCl; 0.5% wt/vol blocking reagent; 0.5% wt/vol BSA) and subsequently incubated overnight in a primary anti-dig-HRP antibody diluted in blocking buffer. On the following day, sections were first washed 3x for 5 min each in TNT buffer (0.1M Tris-HCl, pH 7.5; 0.15M NaCl; 0.3% vol/vol Triton X-100) on a shaker. Secondary antibodies were then applied for probe detection: FITC-tyramide (TSA kit, Perkin Elmer, added as per manufacturer instructions) was applied for 10–15 min to detect dig-HRP; streptavidin conjugated to Alexa 488 was applied for 45 min at room temperature to detect biotin conjugated probes. Sections were then washed 3x for 5 min in TNT buffer on a shaker, counterstained for nuclei with Hoechst 33258 and mounted in a water based mounting solution (Aquamount). All sections were imaged on a Zeiss M1 Axioscope.

## qRT-PCR

Template cDNA was prepared by reverse transcription from 250 ng or 500 ng of total RNA obtained using the Trizol method. Gene expression was quantified using real time quantitative PCR with the SYBR GREEN system (Roche) and primers specific to individual genes. All reactions were performed in technical duplicates or triplicates on an Eppendorf Realplex4 thermocycler (Eppendorf). Ct values were normalized to *Gapdh* expression. For direct comparison of the relative abundance of exons 8 and 8A, standard curves were generated using exon 8- or exon 8A-containing template cDNA and primer efficiency (E) was calculated. Relative abundance was then calculated using the following equation: $(E_{target}^{-Ct\ (target)}) / (E_{GAPDH}^{-Ct\ (GAPDH)})$. The list of primers used is included below:

*mouse Cacna1c-exon8 fwd* 5'-CTGACGGTGTTCCAGTGTATCA-3'
*mouse Cacna1c-exon8 rev* 5'-ACTCATAGCCCATAGCGTCTTG-3'
*mouse Cacna1c-exon8A fwd* 5'-GTCAATGATGCCGTAGGAAG-3'
*mouse Cacna1c-exon8A rev* 5'-CCGCTAAGAACACCGAGAA-3'
mouse Pax6 fwd 5'-TACCAGTGTCTACCAGCCAAT-3'
*mouse Pax6 rev* 5'-TGCACGAGTATGAGGAGGTCT-3'
mouse Nestin fwd 5'-CCCTGAAGTCGAGGAGCTG-3'
*mouse Nestin rev* 5'-CTGCTGCACCTCTAAGCGA-3'
*mouse Gad67 (Gad1) fwd* 5'-CACAGGTCACCCTCGATTTTT-3'
*mouse Gad67 (Gad1) rev* 5'-ACCATCCAACGATCTCTCTCATC-3'
*mouse Gapdh fwd* 5'-AGGTCGGTGTGAACGGATTTG-3'
*mouse Gapdh rev* 5'-TGTAGACCATGTAGTTGAGGTCA-3'
*human CACNA1C-exon8 fwd* 5'-ACGCTATGGGCTATGAGTTACC-3'
*human CACNA1C-exon8 rev* 5'-GGCCTTCTCCCTCTCTTTG-3'
*human CACNA1C-exon8A fwd* 5'-TTTGACAACTTTGCCTTCGC-3'

*human CACNA1C-exon8A rev* 5'-TCCCTTCCTACGGCATCATT-3'
*human NCAM fwd* 5'-ACATCACCTGCTACTTCCTGA-3'
*human NCAM rev* 5'-CTTGGACTCATCTTTCGAGAAGG-3'
*human GAPDH fwd* 5'-CATGAGAAGTATGACAACAGCCT-3'
*human GAPDH fwd* 5'-AGTCCTTCCACGATACCAAAGT-3'

## Fluidigm single-cell dynamic arrays

Embryonic and neonatal C57/bl6 brains (n = 3 mice per time point) were dissociated in papain solution for 5 min at room temperature, followed by the addition of inhibitor solution and gentle trituration to generate single-cell suspensions. Human iPSC-derived neuronal cultures at day 45 of differentiation from TS patients and control subjects (n = 3 lines each for controls and patients) were dissociated in accutase, and FACS sorting was performed at the Stanford shared FACS facility as previously described (*Paşca et al., 2011*; *Ho et al., 2018*), excluding debris, dead cells and doublets to sort single cells into multi-well plates. sc-qPCR experiments were performed as in previous studies (*Paşca et al., 2011*; *Yoo et al., 2011*; *Portmann et al., 2014*) on 96 × 96 dynamic arrays (Fluidigm) and as recommended by the array manufacturer.

## iPSC maintenance and neuronal differentiation

Collection of dermal fibroblasts from TS patients and unaffected control subjects, iPSC generation and maintenance, as well as neuronal differentiation into excitatory cortical projection neurons, were previously described (*Paşca et al., 2011*). All data on neuronal cultures from TS patients and control subjects are from day 45 of neuronal differentiation. All iPSC lines used in this study were previously generated and validated using standard approaches (*Paşca et al., 2011*), and all lines were tested for mycoplasma contamination and maintained mycoplasma free. Informed consent was obtained from all subjects, and the study was approved by the Stanford University IRB panel.

## Immunohistochemistry

Sections were briefly washed with PBS and subsequently permeabilized and blocked for fluorescence immunohistochemistry with 10% normal goat serum in PBS and 0.3% Triton X-100 for 1 hr. Primary antibodies were incubated at 4°C overnight and appropriate fluorophore conjugated Alexa secondary antibodies were applied on the following day for 1 hr at 25°C after three washes with PBS. Following incubation with secondary antibodies, sections were washed three times and subsequently counterstained with the nuclear dye Hoechst 33258 (Molecular Probes, 1:10,000). Sections were then cover slipped with Aqua Poly/Mount (Polysciences) and stored in the dark until imaging. The primary antibodies used included: rabbit anti-Ca$_v$1.2 (Millipore #ab5156, 1:150); chicken anti-GFP (Abcam #ab13970, 1:500); mouse anti-SATB2 (Abcam #ab51502, 1:100); rat anti-CTIP2 (Abcam #ab18465, 1:500); rabbit anti-Tbr1 (Abcam #ab31940, 1:200); mouse anti-6xHis (Abcam #ab5000, 1:100); mouse anti-PAX6 (Developmental Studies Hybridoma Bank, deposited to the DSHB by Kawakami, A., 1:100); rabbit anti-PAX6 (Thermo Fisher, previously Covance #PRB-278P, 1:200); and rat anti-TBR2 (Thermo Fisher #14-4875-82, 1:500). All antibodies used in our study are commercially available and have been previously reported in numerous published studies. We validated all antibodies by negative control experiments omitting the primary antibody, by immunostaining or western blot analyses on null mice, and/or by confirming appropriate localization of immunoreactivity on positive control tissues.

## Protein extraction and western blotting

Cortices from four embryonic C57/bl6 mouse brains from two separate litters were microdissected at E14 and E18 and homogenized in 320 mM sucrose and 20 mM HEPES (pH 7.2). Homogenates were centrifuged at 1000 g for 10 min to eliminate the nuclear fraction and the supernatant was subsequently mixed with 1 vol of 5x Laemmli buffer and stored at −80°C until further analysis. Samples were separated on a Novel NuPAGE 4–12% Bis-Tris gel (Invitrogen) according to manufacterer's instructions and analyzed by western blotting using a rabbit anti-Ca$_v$1.2 antibody (Millipore #ab5156, 1:500) and a mouse anti-GAPDH antibody (Sigma #G8795, 1:1000) as a loading control. Blots were imaged on an Odyssey infra-red scanner (LI-COR) using IRDye 680 and IRDye 800-coupled anti-rabbit and anti-mouse secondary antibodies, respectively (Licor, 1:5000).

## Calcium imaging of primary cortical NPC cultures and transfected Neuro2A cells

We dissociated E12.5-E14.5 cortices into single-cell suspensions and plated them on coverslips in Neurobasal media containing B27 in the presence of bFGF and EGF (20 ng/ml) for 6–24 hr. Primary cultures were loaded with 1 μM of the ratiometric dye Fura-2 acetoxymethyl ester (Invitrogen) for 30 min at 37°C in Neurobasal media containing B27. Cells were then de-esterified in fresh media for 10 min, washed with Tyrode's solution and placed in a Warner Instruments perfusion chamber on an automated stage of an inverted epifluorescence microscope equipped with an excitation filter wheel (TE2000U; Nikon). Solutions were perfused across cells using a manual injection setup. Cells were stimulated with application of 30 μM GABA in a low KCl Tyrode's solution (5 mM KCl, 129 mM NaCl, 2 mM CaCl$_2$, 1 mM MgCl$_2$, 30 mM glucose, and 25 mM HEPES, pH 7.4), in the presence or absence of 5 μM of the LTC inhibitor nimodipine, as delineated in Supplementary *Figure 2*. Consistent with previous reports (*LoTurco et al., 1995*; *Bittman et al., 1997*), only a fraction of cells responded to GABA depolarization with a calcium rise, as many NPCs are gap junction-coupled during early neurogenesis. In a second set of imaging experiments, Neuro2A cells transfected with Ca$_v$1.2 channels and auxiliary subunits were similarly loaded with 1 μM of Fura-2 in Neurobasal media containing B27 for 30 min at 37°C. Cells were then de-esterified in fresh media for 10 min, washed with Tyrode's solution and transferred to the perfusion chamber. Channel-transfected cells were stimulated with application of high KCl Tyrode's solution (67 mM KCl, 67 mM NaCl, 2 mM CaCl$_2$, 1 mM MgCl$_2$, 30 mM glucose and 25 mM HEPES, pH 7.4), as depicted in *Figure 2*. Imaging was performed at room temperature. Openlab software was used to collect time-lapse excitation ratio images and fluorescence images were subsequently analyzed using IGOR Pro software (WaveMetrics).

## Construct generation

Construction of dihydropyridine resistant Ca$_v$1.2 channels containing exon 8 (DHP-Ca$_v$1.2) has been previously described in the pcDNA4/HisMax vector (*Dolmetsch et al., 2001*). Wild-type Ca$_v$1.2 *in utero* over expression constructs were generated by insertion of PCR amplified Ca$_v$1.2 coding sequence from DHP-Ca$_v$1.2 into pCAGIG (a kind gift from Dr. Connie Cepko through Addgene, Addgene plasmid 11159 [*Matsuda and Cepko, 2004*]) using XhoI/NotI restriction sites. pCAGIG-Ca$_v$1.2-TS mutant channels were generated by insertion of PCR amplified Ca$_v$1.2-TS coding sequence from DHP-TS-Ca$_v$1.2 constructs (*Krey et al., 2013*) into pCAGIG. 4EQ pore mutant channels were generated by site directed mutagenesis of four key glutamic acid residues required for calcium selectivity (*Krey et al., 2013*; *Yang et al., 1993*). Mutation of the glutamic acid residues at positions 1087, 2116, 3352 and 4255 into glutamine renders the channels calcium impermeable, as confirmed by ratiometric calcium imaging. All constructs were sequenced through, tested for expression following transfection into HEK293 cells (ATCC), and tested for function using ratiometric calcium imaging following transfection into Neuro2A neuroblastoma cells (ATCC). Ca$_v$1.2-TS-IRES-EGFP, wild-type (WT) Ca$_v$1.2-IRES-EGFP, and pore mutant (4EQ) Ca$_v$1.2-IRES-EGFP constructs encoding Ca$_v$1.2 channels were co-transfected with encoding auxiliary β1β and α2δ1 subunits using lipofectamine 2000 (Thermo Fisher). Transfection of WT constructs induced a robust calcium rise, whereas pore mutant channels showed completely abrogated calcium rises in response to depolarization (*Figure 2d*). Transfection of the TS channel, consistent with a loss of VDI, caused a persistent elevated calcium rise (*Figure 2d*). For *in utero* loss-of-function experiments, pCAG-cre:EGFP was kindly provided by Dr. Connie Cepko through Addgene (Addgene plasmid 13776, *Matsuda and Cepko, 2007*). All plasmids generated for use in this study were purified using the Qiagen EndoFree Maxi Kit.

## *In utero* electroporation

All animal experiments were performed in accordance with the National Institutes of Health, Stanford and UCSF guidelines for the care and use of laboratory animals. For gain-of-function experiments, electroporations were performed on separate randomly assigned animals across multiple litters from timed-pregnant Swiss Webster dams. For loss-of-function experiments, we utilized mice expressing a null allele (*Cacna1c$^-$*) and a conditional allele for Ca$_v$1.2 with loxP sites flanking exons 14 and 15 (*Cacna1c$^{flx}$*). The generation of the conditional null allele (*Cacna1c$^-$*) has been previously

described (*Gomez-Ospina et al., 2013*). Genotyping was performed as in *Gomez-Ospina et al. (2013)*.

In order to introduce specific DNA expression vectors into fetal brains, pregnant mice were first deeply anesthetized using isofluorane inhalation anesthesia. The uterine horns were exposed by a ventral midline incision. Approximately 1 μl of DNA solution (1 μg/μl) was injected through the uterine wall directly into the lateral ventricles of each embryo using a sterile glass micropipette. Once the DNA was injected, two coiled platinum electrodes were placed outside of the uterus, on either side of fetus' head. Using a square pulse electroporator (BTX 830), five 50 msec pulses of 38-40V (at E13.5, 35V at E12.5; 950 msec interval between each pulse) were delivered. Following electroporation of all the embryos, the muscles of the abdominal wall of the pregnant female were sutured and the skin was stapled closed. Mice were closely monitored until fully awake and able to freely ambulate about the cage. Embryos were subsequently harvested for histological analysis at various time points after electroporation.

### Imaging and quantification

All images were captured on a Zeiss M1 Axioscope or Leica Dmi8 microscope. In order to quantify the proportion of electroporated cells that expressed CTIP2, SATB2, or Tbr1 at E16-E17, epifluorescence images were first acquired in the 20x field of view on a Zeiss M1 Axioscope microscope. To quantify the proportion of electroporated cells expressing PAX6 or TBR2 at E15, epifluorescence images were acquired in the 20x field of view on an inverted Leica Dmi8 microscope. Images were post-processed on ImageJ or Adobe Photoshop CS5. Whenever possible, analyses were conducted with the investigator blind to the construct injected. We manually counted the fraction of electroporated cells in the cortical plate (for CTIP2, SATB2, Tbr1) or the VZ/SVZ (for Pax6/Tbr2) that expressed each protein of interest in ImageJ within a counting window of 600 pixels width placed along the lateralmost region of the cerebral wall. Counts encompassed total cells from at least three to six sections per animal spanning caudal to rostral cortex (approximately 2.40 mm to 1.30 mm, according to E17.5 Paxinos developing mouse brain atlas), at least 100 cells per animal, from at least three animals per experimental group. At least two separate litters per experimental group were assessed. For both gain- and loss-of-function electroporation experiments, we assume our analyses incorporate embryos of both sexes at similar proportions. Data are presented as box and whisker plots, with the box bounding the interquartile range (IQR) divided by the mean and whiskers extending to the minimum and maximum value.

### Statistics

Sample sizes were determined empirically based on previous studies in the field, in order to provide sufficient power for statistical comparisons. Statistical analyses were performed on Graphpad Prism v.8 using t test, Mann-Whitney test, Chi-Square test, Kruskal-Wallis test, or ANOVA, with post-hoc correction for multiple comparisons, as noted in the figure legends. Data are presented as box and whisker plots or mean ± SEM, and statistical significance is assumed when $p < 0.05$. Detailed statistical reporting is provided in the corresponding figure legends. Individual data points are shown for all experiments.

## Acknowledgements

We thank K Timothy and the individuals with Timothy syndrome who participated in this study; Steven Sloan and Ben Barres for providing human fetal cortical tissue; Elaine Fisher and Teresa Torres for excellent technical assistance; members of the laboratory of Susan McConnell for providing training in *in utero* electroporation; Thomas Clandinin, Susan McConnell, and Licia Selleri for critical scientific advice and valuable discussions; and the Stanford Flow Cytometry Core Facility for assistance with single cell sorting. This work was supported by an NIMH F31 MH090648 predoctoral fellowship, the Frances B. Nelson Stanford Neuroscience Fellowship, and the UCSF Program for Breakthrough Biomedical Research, Sandler Foundation (to GP); a Howard Hughes Medical Institute international student research award and Lucile P Markey Stanford Graduate Fellowship (to AR); a Swiss National Science Foundation fellowship PBSKP3-123434 (to TP); a NARSAD Young Investigator Award (to SPP); generous support from the Simons Foundation (SFARI 206574), Blume Foundation, and NIMH R01MH096815 (to TDP); and the NIH Pioneer Award 5DP1OD3889 (to RD).

## Additional information

### Competing interests

Thomas Portmann: is currently an employee of Neucyte, Inc. Ricardo E Dolmetsch: is currently an employee of Novartis Institutes for Biomedical Research. The other authors declare that no competing interests exist.

### Funding

| Funder | Grant reference number | Author |
|---|---|---|
| National Institutes of Health | F31 MH090648 Predoctoral Fellowship | Georgia Panagiotakos |
| Stanford University School of Medicine | Frances B Nelson Neuroscience Graduate Fellowship | Georgia Panagiotakos |
| University of California, San Francisco | Program for Breakthrough Biomedical Research, Sandler Foundation | Georgia Panagiotakos |
| Howard Hughes Medical Institute | International Student Research Award | Anshul Rana |
| Stanford University School of Medicine | Lucile P Markey Graduate Fellowship | Anshul Rana |
| Schweizerischer Nationalfonds zur Förderung der Wissenschaftlichen Forschung | Postdoctoral Fellowship PBSKP3-123434 | Thomas Portmann |
| Brain and Behavior Research Foundation | NARSAD Young Investigator Award | Sergiu P Paşca |
| Simons Foundation | SFARI 206574 | Theo D Palmer |
| Blume Foundation | | Theo D Palmer |
| National Institutes of Health | R01 MH096815 | Theo D Palmer |
| National Institutes of Health | Pioneer Award 5DP1OD3889 | Ricardo E Dolmetsch |

The funders had no role in study design, data collection and interpretation, or the decision to submit the work for publication.

### Author contributions

Georgia Panagiotakos, Conceptualization, Data curation, Formal analysis, Supervision, Funding acquisition, Validation, Investigation, Visualization, Methodology, Writing - original draft, Project administration, Writing - review and editing; Christos Haveles, Arpana Arjun, Ralitsa Petrova, Formal analysis, Validation, Investigation, Methodology, Writing - review and editing; Anshul Rana, Thomas Portmann, Sergiu P Paşca, Conceptualization, Formal analysis, Validation, Investigation, Visualization, Methodology, Writing - review and editing; Theo D Palmer, Ricardo E Dolmetsch, Conceptualization, Formal analysis, Supervision, Funding acquisition, Validation, Visualization, Methodology, Project administration, Writing - review and editing

### Author ORCIDs

Georgia Panagiotakos (iD) https://orcid.org/0000-0001-9444-2480
Ralitsa Petrova (iD) https://orcid.org/0000-0001-5586-6192
Sergiu P Paşca (iD) http://orcid.org/0000-0002-3216-3248
Ricardo E Dolmetsch (iD) https://orcid.org/0000-0002-2738-8338

### Ethics

Human subjects: Collection of dermal fibroblasts from TS patients and unaffected control subjects, as well as iPSC generation, was previously described (Pasca, et al. 2011). Experiments involving

primary dermal fibroblasts and iPSCs from Timothy syndrome patients and healthy control subjects were conducted at Stanford University under study protocols (#12481, #232, and #327) approved by the Institutional Review Board and Stem Cell Research Oversight (SCRO) committees of Stanford University after obtaining informed consent.

Animal experimentation: All animal experiments performed in this study were done so in accordance with the National Institutes of Health guidelines for the care and use of laboratory animals and protocols approved by the Stanford University and University of California, San Francisco (UCSF) Institutional Animal Care and Use Committees (Stanford protocol #13705 granted to the lab of Dr. Ricardo Dolmetsch, and UCSF protocol #AN109792 granted to the lab of Dr. Georgia Panagiotakos).

## Decision letter and Author response

Decision letter https://doi.org/10.7554/eLife.51037.sa1
Author response https://doi.org/10.7554/eLife.51037.sa2

## Additional files

### Supplementary files

• Supplementary file 1. Key Resources Table.

• Transparent reporting form

### Data availability

All data analyzed in this study have been included in the manuscript and supporting files and figures. Source data files have been provided for Figures 1 to 4, as well as Figure 1—figure supplement 1.

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
