## [Decision Letter]

**Acceptance summary:**

Your work links calcium channel gene expression in cortical progenitors with sequential development of cortical layer neurons through analysis of the Timothy syndrome (TS) mutation in the alternatively spliced exon (8A) of the calcium channel Ca_v_1.2. Using both mouse and human models to dissect the consequences of the mutation, the results show that the TS mutation prevents a developmental switch in exon expression (8 to 8A) that normally occurs during cortical development. The findings impact both our understanding of the complex sequential processes that are precisely orchestrated during cortical development and the dysfunction that may underlie neurodevelopmental disorders including autism spectrum disorders.

**Decision letter after peer review:**

Thank you for submitting your article "Excess calcium drives defects in differentiation in a syndromic form of autism spectrum disorders" for consideration by *eLife*. Your article has been reviewed by three peer reviewers, including Anita Bhattacharyya as the Reviewing Editor and Reviewer #1, and the evaluation has been overseen by Huda Zoghbi as the Senior Editor. The following individual involved in review of your submission has agreed to reveal their identity: Per Uhlén (Reviewer #2).

The reviewers have discussed the reviews with one another and the Reviewing Editor has drafted this decision to help you prepare a revised submission.

Summary:

The manuscript by Panagiotakos et al., links calcium channel gene expression in cortical progenitors with sequential development of cortical layer neurons. The focus of the manuscript is Timothy syndrome (TS), an autism spectrum disorder affecting multiple organ systems that is caused by a mutation in the alternatively spliced exon (8A) of the calcium channel Ca_v_1.2. The authors show that the TS mutation prevents a developmental switch in exon expression (8 to 8A) that normally occurs during cortical development. Through a series of expression studies, as well as gain and loss of function experiments, the authors show that the lack of exon switching causes altered differentiation of neuronal subtypes, specifically Layer V neurons. The reviewers agree that the authors compellingly show that the expression ratio of exon 8/8A affects neural cell fate during corticogenesis and that the inclusion of both mouse and human experiments is a strength of the work.

Essential revisions:

1) A major concern raised by all reviewers is that the authors have not convincingly shown that altered calcium signaling, as the title states, drives altered differentiation although they have shown that disruption of calcium channel splicing patterns does. The authors rely on previous data and the use of calcium impermeable mutant channels to make the case that the developmental defects are dependent on calcium influx, but excessive calcium in neural progenitors that drives altered differentiation has not been directly shown. This concern can be addressed by modifying the title or by providing more explanation.

2) The authors rely heavily on their previous work but do not provide enough of a discussion of the work to enable the paper to stand alone. The characterization of the effects of the TS-channel mutation on Ca^2+^ dynamics only becomes apparent only when reading the lab's previous work (Pasca et al., 2011). The manuscript would benefit from expanding the text on how TS-mutant channel effects Ca^2+^ dynamics, i.e. the increase in sustained phase, the inactivation dynamics etc. An explanation of what is driving the increased Ca^2+^ load in mutated cells is needed.

3) The authors should clarify discrepancies in the expression of mRNA and protein expression levels in the developing mouse brain. In contrast to mRNA expression, the Ca_v_1.2 protein seems to be expressed at rather low level (if any) in VZ/SVZ at E14 but high in IZ and remains in CP. The WB in Figure 2C also showed a dramatic increase of protein expression at E18 compared to E14, which seems due to the increased exon 8 inclusion (Figure 1A, the total mRNA of 8 + 8a doesn't seem to change that dramatic between E18 and E14). This raises the question if 8A mRNA is actively translated at early stages or the detection of protein at early stage is due to the expression (albeit at low level) of exon 8? Or despite the relatively homogeneous mRNA expression of 8A at E 14 in VZ/SVZ, only a few cells express Ca_v_1.2 protein? If only a small number of progenitor cells express Ca_V_1.2, in utero electroporation of wildtype Ca_v_1.2 sequence may cause gain-of-function phenotype in stem/progenitor cells that do not normally express Ca_V_1.2. This seems to be the case when comparing the number of GFP+ cells that express CTIP2 or SATB2 in Figure 2J and 3D. The mice that received Ca_v_1.2TS-EGFP and Ca_v_1.2WT-EGFP have similar% of CTIP2 and SATB2 positive cells. The authors should add additional discussion to clarify these issues.

4) Reviewers had concerns about the cells' response to calcium in Figure 2. The authors may consider moving the Ca^2+^ results in the supplementary figure to the main figure. In addition, the authors should address the following questions:

What percentage of cells displayed Ca^2+^ rises, how many where blocked, and by how much?

Did cells exhibit spontaneous Ca^2+^ activity, and if so, did it differ between genotypes?

Why did the authors use GABA to trigger a Ca^2+^-response?

Please provide details of how the Ca^2+^ experiments where conducted (i.e. perfusion or bath-application of the drugs that is not described in the Materials and methods section).

---

## [Author Response]

Essential revisions:1) A major concern raised by all reviewers is that the authors have not convincingly shown that altered calcium signaling, as the title states, drives altered differentiation although they have shown that disruption of calcium channel splicing patterns does. The authors rely on previous data and the use of calcium impermeable mutant channels to make the case that the developmental defects are dependent on calcium influx, but excessive calcium in neural progenitors that drives altered differentiation has not been directly shown. This concern can be addressed by modifying the title or by providing more explanation.

We thank the reviewers for making this distinction. We have amended the title to emphasize the splicing change (“Aberrant calcium channel splicing drives cortical differentiation defects in a syndromic autism spectrum disorder”), and we have added additional text to more thoroughly explain the effects of the TS mutation on intracellular calcium elevations upon depolarization. Briefly, the splicing shift observed in patient cells prevents the switch to exon 8 expression and extends the developmental expression of exon 8A – this causes prolonged expression of the TS mutation during differentiation. As the TS mutation is known to prolong voltage-dependent inactivation and thus intracellular calcium rises in response to depolarization, we reasoned that continued expression of the mutation would result in increased calcium elevations during the transition from NPC to post-mitotic neuron. We can see how this may have been unclear in the initial submission, and we believe that the changes to the title and text may address this issue for readers.

2) The authors rely heavily on their previous work but do not provide enough of a discussion of the work to enable the paper to stand alone. The characterization of the effects of the TS-channel mutation on Ca^2+^ dynamics only becomes apparent only when reading the lab's previous work (Pasca et al., 2011). The manuscript would benefit from expanding the text on how TS-mutant channel effects Ca^2+^ dynamics, i.e. the increase in sustained phase, the inactivation dynamics etc. An explanation of what is driving the increased Ca^2+^ load in mutated cells is needed.

We thank the reviewers for this feedback, and we have amended the text of the Introduction and Discussion accordingly to provide a more substantive explanation of how the TS mutation alters Ca_v_1.2-dependent calcium elevations. We have also moved the calcium imaging experiment from Figure 2—figure supplement 1 into main Figure 2 – this experiment, conducted by expressing Ca_v_1.2 channels in a heterologous system, reinforces what we know about how the TS mutation alters intracellular calcium elevations. Consistent with previous work demonstrating that the TS mutation prolongs voltage-dependent channel inactivation, calcium elevations resulting from expression of a TS channel are more sustained as compared to channels not expressing the mutation. Pore-mutant channels, owing to four mutations in the pore-forming region of the channel (4EQ), do not pass calcium upon depolarization.

Reassuringly, the reduction in the relative abundance of SATB2+ cells that we observe with over-expression of channels containing the TS mutation is consistent with gene expression changes we previously reported in neurons generated from TS patients (Pasca et al., 2011). Introduction of wild-type and pore mutant channels further support the idea that changes in calcium elevations may underlie differentiation phenotypes in TS and, coupled with the loss-of-function studies, suggest that tight regulation of calcium levels in differentiating cells may be important for normal cell fate specification.

3) The authors should clarify discrepancies in the expression of mRNA and protein expression levels in the developing mouse brain. In contrast to mRNA expression, the Ca_v_1.2 protein seems to be expressed at rather low level (if any) in VZ/SVZ at E14 but high in IZ and remains in CP. The WB in Figure 2C also showed a dramatic increase of protein expression at E18 compared to E14, which seems due to the increased exon 8 inclusion (Figure 1A, the total mRNA of 8 + 8a doesn't seem to change that dramatic between E18 and E14). This raises the question if 8A mRNA is actively translated at early stages or the detection of protein at early stage is due to the expression (albeit at low level) of exon 8? Or despite the relatively homogeneous mRNA expression of 8A at E 14 in VZ/SVZ, only a few cells express Ca_v_1.2 protein? If only a small number of progenitor cells express Ca_V_1.2,*in utero*electroporation of wildtype Ca_v_1.2 sequence may cause gain-of-function phenotype in stem/progenitor cells that do not normally express Ca_V_1.2. This seems to be the case when comparing the number of GFP+ cells that express CTIP2 or SATB2 in Figure 2J and 3D. The mice that received Ca_v_1.2TS-EGFP and Ca_v_1.2WT-EGFP have similar% of CTIP2 and SATB2 positive cells. The authors should add additional discussion to clarify these issues.

We thank the reviewers for this astute observation, as it is one that we have also considered extensively. We have expanded the discussion to tackle this issue and have also included a raw image of the Ca_v_1.2 channel immunostaining through the VZ/SVZ depicted in Figure 2B.

The expression of Ca_v_1.2 mRNA is in fact rather low in the progenitors of the VZ/SVZ. In our *in situ* experiments, TSA amplification was used to visualize the LNA probes, so care should be taken in interpreting expression level from these data. Previously published single cell RNA sequencing data in the embryonic mouse cortex demonstrates lower expression of Ca_v_1.2 transcripts in progenitor cells as compared to neurons between embryonic days E12 and E15 (Telley L, et al. PMID: 31073041). In general, the detection frequency for Ca_v_1.2 mRNAs is relatively low in both human and mouse single cell RNA seq datasets, further reinforcing that Ca_v_1.2 transcripts are expressed at low levels. Ca_v_1.2 protein expression is similarly lower in NPCs as compared to neurons, as shown in our immunostaining. It also appears, based on our immunostaining data and calcium imaging data, that Ca_v_1.2 expression prior to E14 is restricted to a subset of NPCs. Our immunostaining also suggests that an increasing proportion of progenitors express Ca_v_1.2 protein across development. We suspect, based on how long it would likely take to express functional channels at the membrane in our *in utero*electroporation experiments, that it is likely that gain of function of Ca_v_1.2 occurs as cells exit the cell cycle and we thank the reviewers for this astute observation, as it is one that we have also considered extensively. We have expanded the discussion to tackle this issue and have also included a raw image of the Ca_v_1.2 channel immunostaining through the VZ/SVZ depicted in Figure 2B. The expression of Ca_v_1.2 mRNA is in fact rather low in the progenitors of the VZ/SVZ. In our *in situ* experiments, TSA amplification was used to visualize the LNA probes, so care should be taken in interpreting expression level from these data. Previously published single cell RNA sequencing data in the embryonic mouse cortex demonstrates lower expression of Ca_v_1.2 transcripts in progenitor cells as compared to neurons between embryonic days E12 and E15 (Telley L, et al. PMID: 31073041). In general, the detection frequency for Ca_v_1.2 mRNAs is relatively low in both human and mouse single cell RNA seq datasets, further reinforcing that Ca_v_1.2 transcripts are expressed at low levels. Ca_v_1.2 protein expression is similarly lower in NPCs as compared to neurons, as shown in our immunostaining. It also appears, based on our immunostaining data and calcium imaging data, that Ca_v_1.2 expression prior to E14 is restricted to a subset of NPCs. Our immunostaining also suggests that an increasing proportion of progenitors express Ca_v_1.2 protein across development. We suspect, based on how long it would likely take to express functional channels at the membrane in our*in utero*electroporation experiments, that it is likely that gain of function of Ca_v_1.2 occurs as cells exit the cell cycle and initiate differentiation – nevertheless, we cannot rule out the possibility that part of the gain-of function phenotype resulting from electroporation of wild-type channels results from expressing high levels of Cav1.2 in cells that normally express it at low levels (or not at all in some cases). What this observation does reinforce, however, is the idea that excess calcium is likely behind the changes we see in the relative abundance of CTIP2 and SATB2 cells. We have amended the discussion to include consideration of this possibility.

4) Reviewers had concerns about the cells' response to calcium in Figure 2. The authors may consider moving the Ca^2+^ results in the supplementary figure to the main figure. In addition, the authors should address the following questions:What percentage of cells displayed Ca^2+^ rises, how many where blocked, and by how much?Did cells exhibit spontaneous Ca^2+^ activity, and if so, did it differ between genotypes?Why did the authors use GABA to trigger a Ca^2+^-response?Please provide details of how the Ca^2+^ experiments where conducted (i.e. perfusion or bath-application of the drugs that is not described in the Materials and methods section).

Once again, we thank the reviewers for their insight. As per the request of the reviewers, we have moved the calcium imaging data from Figure 2—figure supplement 1 to Figure 2. We have also added additional details in the Materials and methods section regarding how the calcium experiments were conducted.

As a point of clarification, the two sets of calcium imaging experiments presented in the manuscript had different goals. The data originally presented in Figure 2 (which have now been moved to Figure 2—figure supplement 1) were meant to complement the expression analysis of Ca_v_1.2 protein in Figure 2 panels A-C. Those experiments were not necessarily intended to comprehensively survey the complete repertoire of NPC calcium responses to depolarization, but rather to show that in immature cells that could be depolarized by GABA, functional L-type channels were expressed. We selected GABA to depolarize as it is a more specific manipulation that has previously been shown to depolarize NPCs in the embryonic cortex. However, most likely not all NPCs express GABA receptors, and of those that do, only a fraction express L-type calcium channels at the surface, as depicted in Figure 2B. Briefly, in this set of experiments, we dissociated primary cortical cultures from E12.5-E14.5 wild-type mice and grew those cells in growth factors for 6–24h. The fraction of cells that could be depolarized by GABA ranged between 20 and 55%. To the degree that we were able to detect, we did not observe spontaneous calcium elevations in this setting. This is likely because of the short time between when we extracted the cells and when we recorded their calcium elevations, as well as the fact that we dissociated the cells and did not maintain them as slices. Nevertheless, our findings were consistent with previous reports in rat cortical slices demonstrating that only a fraction of cells respond to GABA application with a calcium rise (Loturco J, et al. PMID: 8845153). Over the course of embryonic neurogenesis, gap junction coupling of NPCs gives way to a progressively increasing proportion of NPCs that can be depolarized by GABA (or glutamate). Of the responding cells, greater than 85% had their calcium rises nearly completely abrogated by application of nimodipine (by 50 seconds after application), suggesting that functional L-type channels are expressed at the membrane in at least a fraction of immature cells in the developing cortex. We have included additional individual imaging traces in Figure 2—figure supplement 1.

In the second set of calcium experiments, we expressed the calcium channels that were electroporated *in vivo* in Neuro2a cells to confirm the effects of the TS mutation, as well as the pore mutations in the 4EQ constructs, on intracellular calcium elevations. Consistent with previous demonstrations that the TS mutation prolongs voltage-dependent channel inactivation, expression of TS channels resulted in more sustained calcium rises as compared to wild-type channels not expressing the mutation. Introducing pore mutations prevented any calcium influx upon depolarization.